# Spiking Transformer-CNN for Event-based Object Detection

## Abstract

Spiking Neural Networks (SNNs) enable energy-efficient computation through event-driven computing and multiplication-free inference, making them well-suited for processing sparse events. Recently, deep Spiking Convolutional Neural Networks (CNNs) have shown energy efficiency advantages on event-based object detection. However, spiking CNNs have been limited to local and single-scale features, making it challenging for them to achieve better detection accuracy. To address this challenge, we propose a hierarchical Spiking Transformer-CNN (i.e., Spike-TransCNN) architecture, which is the first attempt to leverage the global information extraction capabilities of Spiking Transformers and the local information capture abilities of Spiking CNNs for event-based object detection. Technically, we first propose using the Spiking Transformer to extract global features and employ a multi-scale local feature extraction CNN module to complement the Spiking Transformers in local feature extraction. Then, we design intra-stage and inter-stage feature fusion modules to integrate global and multi-scale local features within the network architecture. Experimental results demonstrate that our Spike-TransCNN significantly outperforms existing SNN-based object detectors on the Gen1 dataset, achieving higher detection accuracy (mAP 0.336 vs. 0.321) with lower energy consumption (5.49 mJ vs. 7.26 mJ). Our code can be available in the supplementary materials.

## 1 Introduction

Object detection is essential in computer vision and robotics applications. Nevertheless, conventional cameras operating at fixed frame rates struggle in challenging conditions, such as fast motion, over-exposure, and low light, leading to a significant decline in object detection performance Liu et al. (2020); Sayed & Brostow (2021). Recently, event cameras like DVS Lichtsteiner et al. (2008) and ATIS Posch et al. (2010) have surpassed RGB cameras in dynamic range, temporal resolution, and energy efficiency. These advanced capabilities make them especially well-suited for object detection Peng et al. (2023a;b); Wang et al. (2023a; 2024); Zubic et al. (2024); Yuan et al. (2024); Hamaguchi et al. (2023); Zubić et al. (2023); Tomy et al. (2022) in these challenging scenarios.

Most current event-based object detectors rely on Artificial Neural Networks (ANNs) Perot et al. (2020); Li et al. (2022a), which deliver high performance but come with high computational complexity and energy consumption. In contrast, Spiking Neural Networks (SNNs) Maass (1997); Zhu et al. (2022) present a novel approach inspired by the brain's temporal information processing dynamics. SNNs propagate information through binary spike sequences, allowing for energy-effective computing with event-driven computation and multiplication-free inference. This makes SNNs a more efficient and biologically inspired alternative for event-based object detection.

Early SNN-based object detectors are often derived from existing ANNs through conversion, which introduces several limitations. The most critical problem is that most conversion methods are designed for static images and may not be able to handle sparse temporal event data effectively. This is because these methods focus on approximating the activation patterns of ANNs, often neglecting the spatiotemporal information inherent in event data. Furthermore, conversed models like Spiking-YOLO Kim et al. (2020) require a large number of time steps to match the performance of the original ANN. Although Spike Calibration Li et al. (2022b) can reduce this to hundreds of time steps, its effectiveness still hinges on the quality of the original ANN model.

Directly-trained Spiking Convolutional Neural Networks (CNNs) could be trained with much fewer steps Su et al. (2023), but they primarily focus on local features, which can limit the overall detection performance. For instance, Spiking-DenseNet Cordone et al. (2022) employs a multi-layered approach to process features at various local scales, and SFOD Fan et al. (2024) introduces a fusion mechanism to combine spike features across different local scales. Despite these efforts to optimize spike features across multiple local scales to capture the local features, these Spiking CNNs still face challenges in incorporating global and high-level semantic information, constraining their overall performance.

Transformer architectures have increasingly been integrated into SNNs, such as Spikformer Zhou et al. (2023), Auto-Spikformer Che et al. (2024), and Attention-free Spikformer Wang et al. (2023b). These models have demonstrated superior performance over spiking CNNs in various tasks, primarily due to the Transformer's capability for global attention and parallel computation. However, most current research focuses on classification, with limited exploration of Spiking Transformers in the regression task of object detection. Additionally, recent ANN studies Fang et al. (2022); Chen et al. (2023) suggest that combining the global modeling strengths of Transformers with the local feature extraction capabilities of CNNs can further enhance network performance. Nevertheless, this potential has yet to be thoroughly investigated within the context of SNNs.

To address these challenges, we propose a hierarchical Spiking Transformer-CNN (i.e., Spike-TransCNN), which is a directly-trained deep SNN designed to extract both global and multi-scale local features for event-based object detection. Our model is the first attempt to leverage the global information extraction capabilities of Spiking Transformers and the local information capture abilities of Spiking CNNs. We employ spike-driven token selection to selectively capture tokens, and spike self-attention for holistic perception of spike features. Specifically, we first present a multi-scale local feature extraction module to compensate for the limitations of Spiking Transformers in local feature extraction. Furthermore, we design intra-stage and inter-stage feature fusion modules to integrate global and multi-scale local features within the architecture. The results show that our Spike-TransCNN reduces energy consumption by 4.7× compared to the same ANN architecture. Moreover, it significantly outperforms state-of-the-art methods on the Gen1 dataset, achieving a higher mAP and lower energy consumption.

In summary, the main contributions of this work are:

- We propose Spike-TransCNN, a novel *hierarchical Spiking Transformer-CNN*, which is the first hybrid spiking architecture that combines Spiking CNNs and Spiking Transformers to leverage their complementary strengths, yielding both high-accuracy and energy-efficiency in event-based object detection.

- We present *spike-driven token selection* to select tokens and spike self-attention for global spike feature perception, along with *spiking dilated convolution* for extracting local multi-scale features and optimizing them with *temporal-channel joint attention*.

- We design *intra-stage spike feature fusion and inter-stage spike feature fusion modules* that effectively aggregate features extracted from different architectures and multi-scale features from the event stream to improve object detection performance.

## 2  RELATED WORK

**Event-based Object Detection.** Most event-based object detection methods use ANN approaches, such as RED Perot et al. (2020), ASTMNet Li et al. (2022a), and RVT Gehrig (2023), which demonstrate impressive detection performance but come with high energy consumption. More recently, some works have explored achieving energy-efficient object detection for event data using SNNs. For example, Hybrid-SNN Kugele et al. (2021) combines an SNN backbone for efficient event-based feature extraction with an ANN head for object detection tasks. Spiking-DenseNet Li et al. (2022a) is notable for applying SNNs to event-based object detection using the SSD architecture. A feature pyramid structure Zhang et al. (2023) is introduced to support multi-scale feature extraction. SFOD Fan et al. (2024) introduces a fusion mechanism to combine spike features across different scales. Despite these efforts, Spiking CNNs excel at capturing local features but face challenges in integrating global information, which limits overall detection performance.

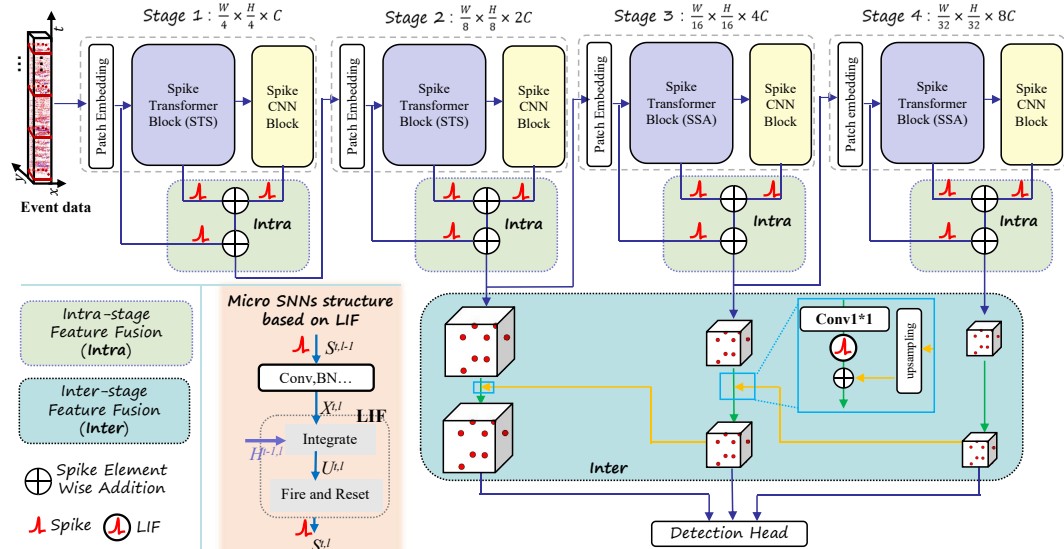

Figure 1: The pipeline of our hierarchical Spiking Transformer-CNNs. Initially, the event stream is processed through a hierarchical hybrid backbone that integrates Spiking Transformer and Spiking CNN blocks. Then, we apply the patch embedding operation across four stages to extract features at four different scales and integrate these features using intra-stage and inter-stage feature fusion modules. Finally, the detection results are predicted on the fused features using the YOLOX head.

**Spiking CNNs.** Current Spiking CNN-based models for object detection can be broadly categorized into two types. The first type involves ANN-to-SNN methods, which convert pre-trained ANNs into SNNs by replacing continuous activation functions with spiking neurons. For example, Spiking-YOLOv4 Wang et al. (2023c) implements a converted CNN model for fast and accurate object detection from event streams. However, these methods face several limitations, including the need for a large number of time steps to match the performance of the original ANN. The second type refers to directly-trained methods, which use surrogate gradients to train deep and large-scale SNNs for object detection. For instance, EMS-YOLO Su et al. (2023) is a directly-trained SNN that surpasses ANN-to-SNN conversion methods, requiring only a few time steps for real-time inference. Besides, a training scheme for SNNs Caccavella et al. (2023) deployed on the neuromorphic chip achieves low-power face detection. These directly-trained SNN models can achieve comparable performance to ANNs with the same architecture while significantly reducing energy consumption.

**Spiking Transformers.** Transformers have been integrated into SNN models with notable success across various tasks. For example, Spikformer Zhou et al. (2023) introduces a pioneering Spiking Self-Attention (SSA) version of the self-attention mechanism. Meanwhile, Spike-Driven Self-Attention (SDSA) Yao et al. (2024) employs mask and addition operations to avoid multiplication, significantly reducing computational energy compared to conventional self-attention mechanisms. Besides, a spatiotemporal self-attention mechanism Wang et al. (2023d) has been proposed for SNNs, effectively capturing feature dependencies while preserving the asynchronous transmission property of SNNs. QKFormer Zhou et al. (2024) utilizes sparse matrices to filter tokens and channels. Despite these advancements in Transformer-based SNN models, the exploration of object detection tasks using event data remains relatively limited.

## 3 METHODS

### 3.1 OVERVIEW

This work aims at designing a novel hierarchical Spiking Transformer-CNN, termed **Spike-TransCNN**, which combines Spiking CNNs and Spiking Transformers to leverage their complementary strengths for high-accuracy and energy-efficient object detection. As depicted in Fig. 1, our framework consists of four key parts: Spiking Transformer Block (STB), Spiking CNN Block

(SCB), intra-state spike feature fusion module, and inter-stage spike feature fusion module. More precisely, our Spike-TransCNN starts by processing the event stream through a hybrid backbone that combines Spiking Transformers and Spiking CNNs. We then extract features at four different scales using patch embeddings (PE) across four stages in the hybrid architecture, integrating these features with both intra-stage and inter-stage fusion modules. Finally, we use the YOLOX head to predict detection results from the fused features.

## 3.2 THE BASICS OF SPIKE NEURAL NETWORKS

Spike Neural Networks (SNNs) are inspired by the brain's functioning, communicating through discrete spikes, enabling efficient information processing for event data. Compared to traditional ANNs, SNNs excel in spatio-temporal processing and energy efficiency. Spiking neurons are the essential components of SNNs, communicating through discrete spikes that mimic the behavior of biological brain neurons. The Leaky Integrate-and-Fire (LIF) neuron model Delorme et al. (1999) is commonly used in SNNs as follows:

$$V_t = V_{t-1} + \frac{1}{\tau}(-(V_{t-1} - V_{rest}) + X_t), \tag{1}$$

$$S_t = Hea(V_t - V_{th}), \tag{2}$$

where $V_t$ denotes the membrane potential after neuronal dynamics at timestep $t$. $X_t$ represents the input to neuron. $\tau$ is the decay factor for leakage. $Hea()$ is the Heaviside step function, which satisfies: $Hea(x) = 1$ when $x \geq 0$, otherwise $Hea(x) = 0$. The generation of output spikes is controlled by the threshold $V_{th}$, and once the neuron emits a spike at time step $t + 1$, the current membrane potential will be reset to $V_{rest}$. SNNs Wu et al. (2018) based on spike neurons (LIF) can be developed for network training as:

$$U^{t,l} = H^{t-1,l} + X^{t,l}, \ S^{t,l} = Hea(U^{t,l} - u_{th}), \tag{3}$$

$$H^{t,l} = V_{rest}S^{t,l} + (\beta U^{t,l}) \odot (1 - S^{t,l}), \tag{4}$$

where $t$ and $l$ respectively represent timestamps and network layers. $V$ represents the membrane potential generated by integrating spatial dimension input $X^{t,l}$ and temporal dimension input $H^{t-1,l}$. $u_{th}$ is the threshold that determines whether the output spike tensor $S^{t,l}$ should be fired or kept as zero. $H^{t,l}$ represents the internal state of neurons propagating over time, where $\beta = e^{\frac{-dt}{\tau}}$ reflects the decay factor, and $\odot$ denotes element-wise multiplication.

## 3.3 SPIKING TRANSFORMERS FOR GLOBAL FEATURES

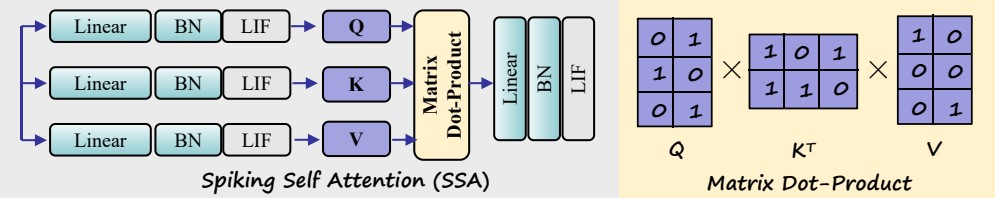

Figure 2: Spiking Self Attention (SSA). SSA is employed for global feature extraction in the latter two stages. The right illustrates the operational processes of 'Matrix Dot-Product'.

To obtain hierarchical features, the patch embedding operation is implemented for downsampling before each STB. It specifically includes conv1 ∗ 1, max-pooling, and LIF. We use Spike-driven Token Selection (STS) to select tokens for sparse features in the first two STB modules (see Fig. 3) and employ Spiking Self-Attention (SSA) to extract global features in the latter two STB modules (see Fig. 2). SSA utilizes sparse spike-form $Q, K, V$ for computation, without the need for softmax operations and floating-point matrix multiplication. The calculation process of SSA can be formulated as follows:

Figure 3: Spike-driven Token Selection (STS). STS is used for token selection in the first two stages.

$$I = SN_I(BN(XW_I)), I \in (Q, K, V), \tag{5}$$

$$SSA(Q, K, V) = SN(BN(Linear(SN(QK^TV * s)))), \tag{6}$$

where $Q, K, V \in R^{T \times N \times D}$, $N$ is the token number, and $D$ is the channel number. The spike-form $Q, K, V$ are computed by learnable linear layers. $s$ represents a scaling factor, and $SN$ refers to the LIF layer.

In order to select important regions within shallow features, we designed Spike-driven Token Selection (STS) in the first two STBs. Firstly, use linear functions to learn the weights $W_v$ and $W_k$ of the value and key domains for each token. Then, utilize LIF to map the key values to (0 or 1) as the gating control for tokens. The STS can be formulated as follows:

$$V = SN_V(BN(XW_V)), K = SN_K(BN(XW_K)), \tag{7}$$

$$G_t = \sum_{i=0}^{D} SN(K_{i,j}), X^{'} = G_t \otimes V, \tag{8}$$

where $G_t$ is the $N * 1$ token attention vector, which models the binary importance of different tokens. D is the channel number. $\otimes$ is the Hadamard product between spike tensors.

In our backbone network, we utilize STS for token filtering in the first two stages, and SSA in the latter two stages. Experimental results demonstrate that the combination of these two attention mechanisms may enhance the detection performance of the network.

## 3.4 SPIKING CNNs FOR MULTI-SCALE LOCAL FEATURES

The STB excels at extracting global information but lacks the ability to capture local details. Therefore, we designed a local multi-receptive field Spiking CNN Block (SCB) that leverages the local feature extraction capabilities of CNNs and automatically selects important local multi-scale features using time channel attention (see Fig. 4).

Initially, the SCB utilizes dilated convolutions with dilation rates of [1, 3, 5] to capture local multi-scale information. Subsequently, the multi-scale local spiking features are stacked along the channel dimension to ensure feature binarization, which can be formulated as follows:

$$X_i = SN_i(BN(DC_{dr=i}(X))), i \in (1, 3, 5), \tag{9}$$

$$X_{SDC} = Cat(X, X_1, X_3, X_5), \tag{10}$$

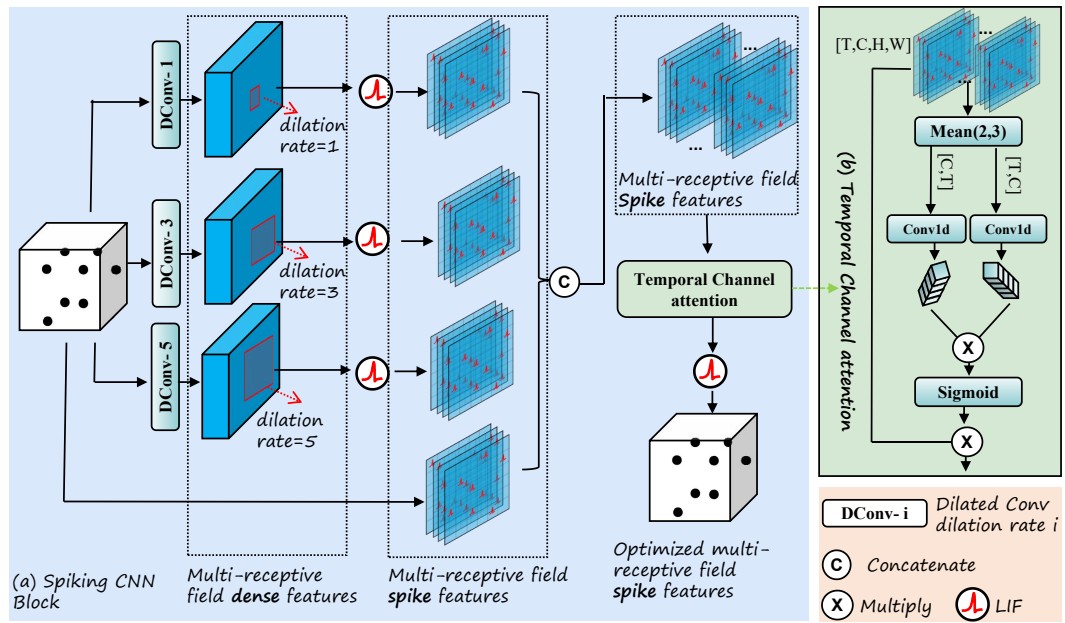

Figure 4: Spiking CNN Block. Extract local multi-receptive field features using dilated convolutions, and select features using temporal channel attention.

where $DC_{dr=i}$ denotes dilated convolution with a dilation rate of $i$, while $Cat$ represents channelwise concatenation.

For the [T,C,H,W] dimensional spiking features $X_{SDC}$, a time channel attention mechanism is employed to automatically select the time and channel of the multi-scale local spiking features. Finally, LIF is used to activate the features, ensuring the binarization of the features input to the next stage and maintaining the binarized characteristics of the SNN network, which can be formulated as:

$$X_{SCB} = SN(X_{SDC} \odot \sigma(Conv1d(F_t) \cdot Conv1d(F_c))), \tag{11}$$

where $F_t$ is the mean of $X_{SDC}$ along dimensions $H$ and $W$, and $F_c$ is obtained by swapping the dimensions (T, C) of $F_t$.

## 3.5 INTRA-STAGE SPIKE FEATURE FUSION

To ensure that the features in each stage used for object detection contain both global features and local multi-receptive field features, we use spike-element-wise (SEW) addition to fuse the original input features, STB output features, and SCB output features in the intra-stage spike feature fusion module. SEW-ResNet Fang et al. (2021) has demonstrated that spike-element-wise addition in residual connections can effectively prevent issues of gradient vanishing and gradient explosion. Fig. 5 illustrates the (a) SEW block Fang et al. (2021) and (b) the proposed intra-stage spike feature fusion module. The specific operations of the intra-stage spike feature fusion module are as follows:

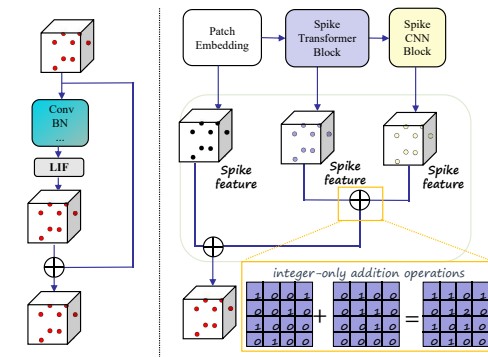

(a) Spike-Element-Wise Block   (b) Spike-Element-Wise Intra-stage Feature Fusion Block

Figure 5: Intra-stage Feature Fusion. (a) SEW Block. (b) Intra-stage spike feature fusion block.

$$X_{Intra} = X_{PE} \oplus (X_{STB} \oplus X_{SCB}), \tag{12}$$

where $X_{PE}$, $X_{STB}$, and $X_{SCB}$ denote the spike features output by PE, STB, and SCB, respectively. $\oplus$ refers to the spike-element-wise addition operation, which ensures that the fusion operation in-

volves only integer addition. The intra-stage spike feature fusion not only ensures the integration of global and local features at each stage, but it also prevents the loss of features extracted by distant STBs when passing input to the next stage.

### 3.6 INTER-STAGE SPIKE FEATURE FUSION

Intra-stage spike feature fusion can merge features with the same spatial resolution, but it lacks interaction between features of different stages. Therefore, we design an inter-stage spike feature fusion module to facilitate interaction between different stages. The specific design of the inter-layer feature fusion module is shown in Fig. 6. It utilizes a $1 \times 1$ convolution to adjust the channel dimensions of the features at the current stage, applies up-sampling to the features from the previous stage, and then adds them element-wise. To ensure the binary and sparse nature of the features, LIF is used for spike activation after each convolution. The operations between two adjacent stages can be formulated as follows:

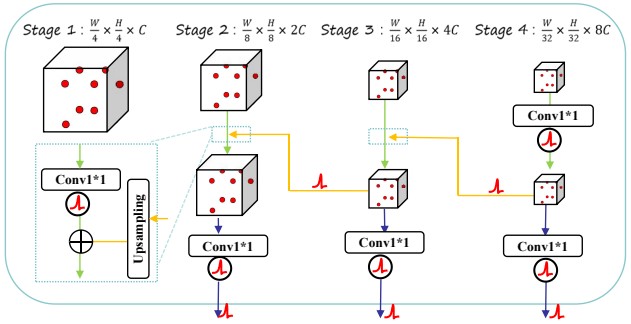

Figure 6: Inter-stage Feature Fusion. This module consists of two main operations: vertical channel modification and horizontal upsampling, enabling interaction between shallow features and deep features over longer distances.

$$X_{S-(j)} = SN(Conv(X_{Intra-(j)})), \tag{13}$$

$$X_{Inter-(j)} = X_{S-(j)} \oplus Up2(X_{S-(j-1)}), \tag{14}$$

$$X_{Inter-(j)} = SN(Conv(X_{Inter-(j)})), \tag{15}$$

where $X_{Intra-(j)}$ denotes the output spike features of the j-th stage after intra-stage feature fusion, and $Up2$ represents 2x up-sampling. This module enables interaction and fusion of features at different depths and resolutions.

Finally, the top three features obtained from the inter-stage feature fusion are fed into the YOLOX Ge et al. (2021) detection head for classification and regression predictions.

## 4 EXPERIMENTS

### 4.1 EXPERIMENT SETTINGS

**Datasets.** We evaluate the effectiveness of our Spike-TransCNN using two publicly available annotated datasets: Gen1 De Tournemire et al. (2020) and 1Mpx Perot et al. (2020). Both datasets were captured by event cameras in real-world driving scenarios. The Gen1 dataset includes 39 hours of event data with a resolution of 304 × 240, providing 228k car and 28k pedestrian bounding boxes labeled at frequencies of 1, 2, or 4 Hz. The 1Mpx dataset offers recordings with a resolution of 720 × 1280, totaling around 15 hours of event data, and includes 25 million bounding box labels at either 30 or 60 Hz for cars, pedestrians, and two-wheelers. For processing these datasets, we follow the methodology outlined in RVT-B Gehrig (2023).

**Implementation Details.** Our models are trained for 400k iterations using the ADAM optimizer Kinga et al. (2015) with a OneCycle learning rate schedule Smith & Topin (2019), which includes 2000 warmup iterations followed by linear decay of the maximum learning rate. This training strategy is consistent across all studies. For the Gen1 dataset, we use a batch size of 4 and a maximum

Table 1: Comparison with state-of-the-art methods and our Spike-TranCNN on the Gen1 dataset and the 1Mpx dataset. Note that, our Spike-TransCNN significantly outperforms existing SNN-based object detectors on the Gen1 dataset.

| Method | Type | T | Params | Firing Rate | Energy (mJ) | Gen1 mAP | 1Mpx mAP |
|---|---|---|---|---|---|---|---|
| Inception Iacono et al. (2018) | CNN | – | >60M | – | – | 0.301 | 0.340 |
| RRC-Events Chen (2018) | CNN | – | >100M | – | – | 0.312 | 0.343 |
| Matrix Cannici et al. (2019) | RNN+CNN | – | 61.5M | – | – | 0.310 | – |
| YOLOv3E Jiang et al. (2019) | CNN | – | >60M | – | – | 0.312 | 0.346 |
| RED Perot et al. (2020) | CNN+RNN | – | 24.1M | – | > 24 | 0.400 | 0.430 |
| ASTMNet Li et al. (2022a) | CNN+RNN | – | >100M | – | – | 0.467 | **0.483** |
| RVT-B Gehrig (2023) | Transformer | – | 18.5M | – | – | **0.472** | 0.474 |
| VGG-11 Li et al. (2022a) | SNN | 5 | 12.64M | 22.22% | 11.06 | 0.174 | – |
| MobileNet Li et al. (2022a) | SNN | 5 | 24.26M | 29.44% | 5.76 | 0.147 | – |
| DenseNet Li et al. (2022a) | SNN | 5 | 8.2M | 37.2% | 3.89 | 0.189 | – |
| FPDAGNet Zhang et al. (2023) | SNN | 5 | 22M | 19.1% | – | 0.223 | – |
| SNN-CN Bodden et al. (2024) | SNN | 5 | 12.97M | 10.8% | – | 0.223 | – |
| KD-CN Bodden et al. (2024) | SNN | 5 | 12.97M | 17.4% | – | 0.229 | – |
| EMS-YOLO Su et al. (2023) | SNN | 5 | 6.20M | 21.15% | – | 0.267 | – |
| EMS-YOLO Su et al. (2023) | SNN | 5 | 9.34M | 20.09% | – | 0.286 | – |
| EMS-YOLO Su et al. (2023) | SNN | 5 | 14.4M | 17.80% | – | 0.310 | – |
| SFOD Fan et al. (2024) | SNN | 5 | 11.90M | 24.04% | 7.26 | 0.321 | – |
| **Spike-TransCNN (ours)** | SNN | 5 | 24.3M | 19.83% | 5.49 | **0.336** | **0.250** |

learning rate of $2e - 4$, running the training on a single NVIDIA A100 GPU. For the 1Mpx dataset, we set an effective batch size of 4, a maximum learning rate of $2.45e - 4$, and also train the model using a single NVIDIA A100 GPU.

## 4.2 MAIN TEST

**Quantitative Evaluation.** As shown in Table 1, we compare state-of-the-art event-based object detection methods with our Spike-TransCNN on the Gen1 dataset and the 1Mpx dataset. While ANN-based models demonstrate high performance, they come with significant energy consumption. Note that, our Spike-TransCNN significantly outperforms ten state-of-the-art methods. More precisely, our Spike-TransCNN surpasses the best competitor, namely SFOD, achieving a higher mAP (0.336 vs. 0.321) and lower energy consumption (5.49 mJ vs. 7.26 mJ). Additionally, it's important to note that most event-based object detectors on the 1Mpx dataset are ANN-based models, with very few SNN-based models. However, our Spike-TransCNN has been tested on 1Mpx to establish a benchmark for future comparisons. The limited use of

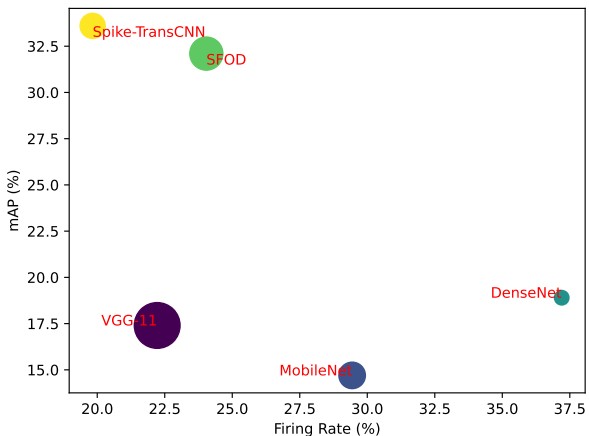

Figure 7: Detection performance vs firing rate of our Spike-TransCNN on the GEN1 dataset. The areas of the circles correspond to the energy.

SNNs on 1Mpx may be attributed to the large dataset size, as SNNs typically require more training resources and video memory than ANNs. Fig. 7 provides a visual comparison of the accuracy, spike firing rate, and energy consumption of our approach with other SNN-based methods.

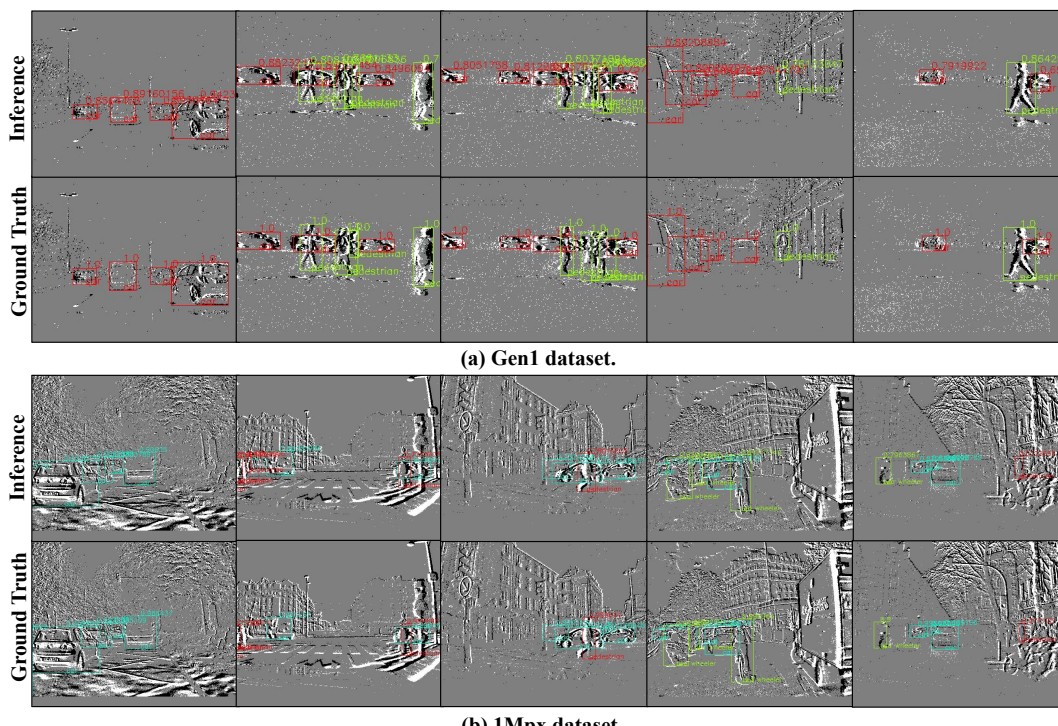

(a) Gen1 dataset.

(b) 1Mpx dataset.

Figure 8: Representative visualization examples of object detection results on the Gen1 dataset and the 1Mpx dataset.

Table 2: The contribution of each component to our Spike-TransCNN on the Gen1 dataset.

| Method | STB | SCB | Intra-stage fusion | Inter-stage fusion | Params | mAP$_L$ | mAP$_M$ | mAP$_S$ | mAP |
|--------|-----|-----|--------------------|--------------------|--------|---------|---------|---------|-----|
| (a) | ✓ | | | | 10.8M | 0.208 | 0.254 | 0.08 | 0.184 |
| (b) | ✓ | ✓ | | | 21.9M | 0.239 | 0.324 | 0.134 | 0.240 |
| (c) | ✓ | ✓ | ✓ | | 23.5M | 0.213 | 0.326 | 0.140 | 0.245 |
| (d) | ✓ | ✓ | ✓ | ✓ | 24.1M | **0.251** | **0.378** | **0.179** | **0.285** |

**Visualization Evaluation.** We further present representative visualization results on the Gen1 dataset and the 1Mpx dataset (Fig. 8). We select challenging scenarios with occlusions and multi-scale objects. The detection results from our Spike-TransCNN are very close to the ground truth. It indicates that our method performs well in some specific scenarios with single-modal input, particularly when deployed on energy-constrained edge devices.

### 4.3 ABLATION EXPERIMENTS

We conduct ablation studies on the test set of the Gen1 dataset. We evaluate the impact of various modules and take a deep look at the impact of each design choice as follows.

**Contribution of Each Component.** To explore the impact of each component on the final performance, we choose Spiking Transformer as the baseline. As shown in Table 2, four methods, namely (a), (b), (d), and (d), utilize Spiking Transformer Block (STB), Spiking CNN Block (SCB), Intra-stage feature fusion module, and Inter-stage feature fusion module, consistently achieve higher performance on the Gen1 dataset than the baseline. Specifically, comparing (b) and (d), the absolute promotion of mAP is 4.5%, which demonstrates that it is feasible to adopt intra-stage spike feature fusion between Spiking Transformer block and Spiking CNN block as well as inter-layer spike feature fusion.

**Influence of Spike-driven Token Selection.** As shown in Table 3, we explore the influence of the multi-head self-attention operation in our Spike-TransCNN on the Gen1 dataset. We can find that replacing SSA with STS in the first two stages and combining STS with SSA enhanced the overall performance of the model.

Table 3: The influence of Spike-driven Token Selection (STS) on the Gen1 dataset.

| Operation | Params | mAP$_L$ | mAP$_M$ | mAP$_S$ | mAP |
|-----------|--------|---------|---------|---------|-----|
| SSA | 24.1M | 0.251 | 0.378 | 0.179 | 0.285 |
| STS+SSA | 24.3M | **0.292** | **0.426** | **0.237** | **0.336** |

## 4.4 ENERGY CONSUMPTION

The energy consumption of SNNs in neuromorphic hardware are usually assessed based on the number of computational operations Su et al. (2023). In ANNs, each operation involves floating-point multiplications and additions (MAC), and the computation cost is estimated using the number of floating-point operations (FLOPs). SNNs exhibit high energy efficiency in neuromorphic hardware because only neurons involved in spike generation contribute to accumulation calculations (AC), and computations can be performed with roughly the same number of synaptic operations (SyOPs). Hence, we quantify the energy consumption of the original SNN as $E_{SNN} = \sum E_l$, where the energy of the $l$-th layer can be calculated as:

Table 4: Energy consumption analysis on the Gen1 dataset.

| Model | #OP$_{AC}$ | #OP$_{MAC}$ | Energy | Efficiency |
|-------|-----------|-------------|--------|------------|
| TransCNN (ANN) | / | 5.59G | 25.70mJ | 1× |
| Spike-TransCNN | 5.45G | 0.14G | 5.49mJ | 4.7× |

$$E_l = T \times (S_{fr} \times E_{AC} \times OP_{AC}) + E_{MAC} \times OP_{MAC}, \tag{16}$$

where $T$ represents the time step, $S_{fr}$ denotes the firing rate, and $OP_{AC}$ and $OP_{MAC}$ represent the numbers of AC and MAC operations, respectively. Table 4 presents the energy consumption of our method compared to the theoretical energy consumption of the ANN with the same network architecture. We assume a 32-bit floating-point implementation using 45nm technology, with energy values of $E_{MAC} = 4.6$ pJ and $E_{AC} = 0.9$ pJ Horowitz (2014). Our Spike-TransCNN has an energy consumption of only 5.84 mJ, achieving a 4.4× improvement in energy efficiency compared to the same ANN architecture.

## 4.5 DISCUSSION

Indeed, our Spike-TransCNN achieves higher detection accuracy and lower power consumption compared to existing pure SNN-based object detection methods, making it suitable for energy-constrained edge devices. Nevertheless, pure SNN models may still exhibit a slight performance gap compared to equivalent ANN architectures or hybrid SNN-ANN models Yu et al. (2024). To further match ANN-level accuracy, we could increase the simulation time steps Luo et al. (2024)or extend the model to handle multiple modalities, such as combining RGB frames with event data.

## 5 CONCLUSION

This paper proposes a novel hybrid network that takes advantage of both Spiking Transformers and Spiking CNNs for event-based object detection. To the best of our knowledge, this is the first use of a hierarchical hybrid network that includes intra-stage and inter-stage spike feature fusion modules to ensure comprehensive integration of global and multi-scale local information. Experimental results demonstrate that our Spike-TransCNN significantly outperforms existing SNN-based object detectors on the Gen1 dataset, achieving higher mAP and lower energy consumption. We believe our work presents a conceptual hybrid framework that integrates Spiking Transformers and Spiking CNNs, offering potential for various event-based vision applications and feasibility for deployment on neuromorphic hardware.

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

# A APPENDIX

## A.1 EVENT DATA PREPROCESSING

The output of an event camera with a resolution of $H \times W$ can be represented as an event sequence, denoted as $E = e_i 1^N$, where $e_i = (x_i, y_i, t_i, p_i)$. Here, $p_i \in -1, 1$ represents the polarity of a brightness change that occurs at time $t_i$ and pixel position $(x_i, y_i)$. The change is triggered for the pixel $(x_n, y_n)$ at timestamp $t_n$ when the log-intensity $\ln L$ changes beyond the pre-defined threshold $\theta$. This dynamic visual sensing mechanism is depicted by the inequality:

$$\ln L(x_n, y_n, t_n) - \ln L(x_n, y_n, t_n - \Delta t_n) p_n \theta, \tag{17}$$

Here, the polarity $p_n \in \{1, -1\}$ indicates whether the brightness is increasing or decreasing, and $\Delta t_n$ represents the temporal sampling interval of DVS at a pixel.

In our investigation, we have implemented the technique presented in Gehrig (2023) for preprocessing event data. Our preprocessing generates a 4-dimensional tensor $E$ from discrete event data. The first dimension consists of T components associated with T discretization steps of time. The second dimension includes two components signifying polarity. The third and fourth dimensions represent the height and width of the event camera. We process a set of events $\varepsilon$ within the time duration $[t_a, t_b)$ in the following manner:

$$E(\tau, p, x, y) = \sum_{\varepsilon} \delta(p - p_n)\delta(x - x_n)\delta(\tau - \tau_n),$$
$$\tau_n = \left\lfloor \frac{t_n - t_a}{t_b - t_a} \cdot T \right\rfloor \tag{18}$$

The provided equation describes the handling of a set of events $\varepsilon$ over a time interval $[t_a, t_b)$, with each $t_n$ falling between $t_a$ and $t_b$. Here, $\delta(\cdot)$ represents the Dirac delta function, where $\delta(t)$ is 0 for all $t \neq 0$, and $\int \delta(t)dt$ equals 1. The value of $T$ is determined by the selected number of discrete time steps. Following this procedure, a four-dimensional tensor $E \in [T, 2, H, W]$ is obtained, where $T$, $H$, and $W$ denote the aggregation time, preprocessed height, and width, respectively.

The datasets used are Gen1 [1] and 1Mpx [2].

## A.2 EVALUATION CRITERIA

Table 5: Summary of common metrics for event-based object detection.

| Metric | Unit | Description |
|---|---|---|
| mAP | - | mAP averaged over ten IoUs: $\{0.5{:}0.05{:}0.95\}$. |
| $mAP_{0.5}$ | - | mAP at a fixed IoU=0.50. |
| $mAP_{0.75}$ | - | mAP at a fixed IoU=0.75. |
| $mAP_S$ | - | mAP for small objects of area smaller than $32^2$. |
| $mAP_M$ | - | mAP for objects of area between $32^2$ and $96^2$. |
| $mAP_L$ | - | mAP for large objects of area bigger than $96^2$. |
| Model size | MB | The number of parameters for the learning-based model. |
| Power consum. | mJ | The energy consumption of the SNN model through AC and MAC operations in a neuromorphic chip. |

The evaluation metrics for object detection based on event data are summarized in Table 5. For both datasets, the primary metric we use is mean average precision (mAP) Lin et al. (2014). Additionally, to demonstrate the capability of our method for detecting objects at multiple scales, we also use $mAP_L$, $mAP_M$, and $mAP_S$. The AP is derived from precision and recall using the following formulas:

$$\text{AP} = \int_0^1 \max\{p(r'|r' \geq r)\}dr, \tag{19}$$

---

[1] https://www.prophesee.ai/2020/01/24/prophesee-gen1-automotive-detection-dataset

[2] https://www.prophesee.ai/2020/11/24/automotive-megapixel-event-based-dataset/

where $r$ denotes the recall, $p(r)$ is the precision-recall curve.

Thus, the mAP is calculated as the average of AP values across all object categories as follows:

$$\text{mAP} = \frac{\sum_{i=1}^{C_b} AP(i)}{C_b}, \tag{20}$$

where $C_b$ represents the number of object classes. At present, MS COCO[3] is the most widely used benchmark for evaluating event-based object detection methods. Instead of using a fixed IoU threshold, MS COCO Perot et al. (2020) provides a few metrics with various IoUs (i.e., mAP, $\text{mAP}_{0.5}$, and $\text{mAP}_{0.75}$) and AP across different scales (i.e., $\text{mAP}_S$, $\text{mAP}_M$, and $\text{mAP}_L$). As neuromorphic cameras offer continuous visual streams, object detection labels are annotated on the RGB image or reconstructed image at a fixed frame rate. Thus, many existing methods assess the detection accuracy at the timestamp when the label is provided, lacking the ability to evaluate the entire event stream continuously.

We calculate the spike Firing Rate by counting the number of spikes and the number of neurons in each layer of the network, which can be computed using the following formula:

$$\text{Firing Rate} = \frac{\text{Numbers of spikes}}{\text{Numbers of neurons}}. \tag{21}$$

## A.3 THE IMPLEMENTATION OF LIF NEURONS

The firing and membrane potential updates are two main modules in the LIF neuron Delorme et al. (1999), for which we provided the implementation functions in Algorithm 1 and Algorithm 2. In Algorithm 1, the forward propagation function during the firing process is presented, and a gradient substitution function is defined in the backward propagation function. We use the functions defined in Algorithm 2 as the LIF neuron for this paper, where the input x has dimensions of [T, 2, H, W]. Table 6 provides specific numerical values of the hyperparameters used in the LIF model in this work.

Table 6: Values of LIF parameters.

| Parameter | thresh | lens | decay | time_window |
|-----------|--------|------|-------|-------------|
| Value | 0.5 | 0.5 | 0.25 | 5 |

## A.4 PATCH EMBEDDING

In order to obtain hierarchical features, we implement the patch embedding operation for downsampling before each STB. This operation consists of $\text{conv}1 * 1$, max-pooling, and LIF. We downsample the spatial resolution using max pooling, which helps preserve the binary nature of the elements.

## A.5 SPIKE TRANSFORMER BLOCK

As shown in Fig. 10, detailed design details of the Spike Transformer Block(STB) are provided. The main difference between the first two stages and the last two stages is the multi-head attention. In the first two stages, we used Spike-driven token selection (STS) in the multi-head attention, and Spike self-attention (SSA) mechanism in the last two stages.

## A.6 THE ORDER OF STB AND SCB

Spiking Transformer Block (STB) and Spiking CNN Block (SCB) serve as the core modules for feature extraction in the backbone. While STB focuses on extracting global features, SCB specializes in capturing local multi-scale features. In each stage of our proposed method, we first employ STB to extract global features, followed by utilizing SCB to capture local multi-scale features. Our analysis of the sequential order of these two modules in each stage, illustrated in Fig. 11, revealed

---

[3]https://github.com/cocodataset/cocoapi

---

**Algorithm 1:** Approximate Firing Function

---

**Input:** $input$
**Output:** $output$

1 **Function** `ActFun`$(ctx, input)$:
    **Data:** $input$
    **Result:** $output$

2   **Function** `forward`$(ctx, input)$:
      **Save** $input$ for backward pass in $ctx$;
      **return** $float(input > thresh)$;

3   **Function** `backward`$(ctx, grad\_output)$:
      **Data:** $grad\_output$
      **Result:** $grad\_input$
      $input \leftarrow$ **Retrieve** saved tensors from $ctx$;
      $grad\_input \leftarrow grad\_output.clone()$;
      $temp \leftarrow abs(input - thresh) < lens$;
      $temp \leftarrow temp/(2 \times lens)$;
      **return** $grad\_input \times float(temp)$;

    **Invoke** forward and backward functions as needed;
    **return** *forward(ctx, input)*;

---

**Algorithm 2:** Membrane Potential Update

---

**Input:** $x$
**Output:** $output$

1 **Function** `mem_update`$(x)$:
    **Data:** $x$
    **Result:** $output$

2   $mem \leftarrow torch.zeros\_like(x[0]).to(device)$;
3   $spike \leftarrow torch.zeros\_like(x[0]).to(device)$;
4   $output \leftarrow torch.zeros\_like(x)$;
5   $mem\_old \leftarrow 0$;
6   **for** $i = 0$ **to** $time\_window - 1$ **do**
7     **if** $i \geq 1$ **then**
8       $mem \leftarrow mem\_old \times decay \times (1 - spike.detach()) + x[i]$;
9     **else**
10       $mem \leftarrow x[i]$;
11     $spike \leftarrow ActFun(mem)$;
12     $mem\_old \leftarrow mem.clone()$;
13     $output[i] \leftarrow spike$;
14   **return** $output$;

---

that the model is minimally impacted by the order in which they are applied. As depicted in Table 7, regardless of whether SCB is used before STB (as in A) or vice versa (as in B), the overall effect on the model is negligible. We attribute this to the fusion module proposed in this paper, which comprehensively integrates global and local features. Notably, the information from the preceding module is retained, irrespective of the order in which STB and SCB are employed.

## A.7 SPIKE-ELEMENT-WISE ADDITION

In this study, $\oplus$ operations are employed in both intra-stage and inter-stage contexts. Herein, we aim to demonstrate how this operation effectively mitigates issues such as gradient vanishing or gradient explosion.

Taking the SEW Block Fang et al. (2021) as an example, upon the application of $\oplus$:

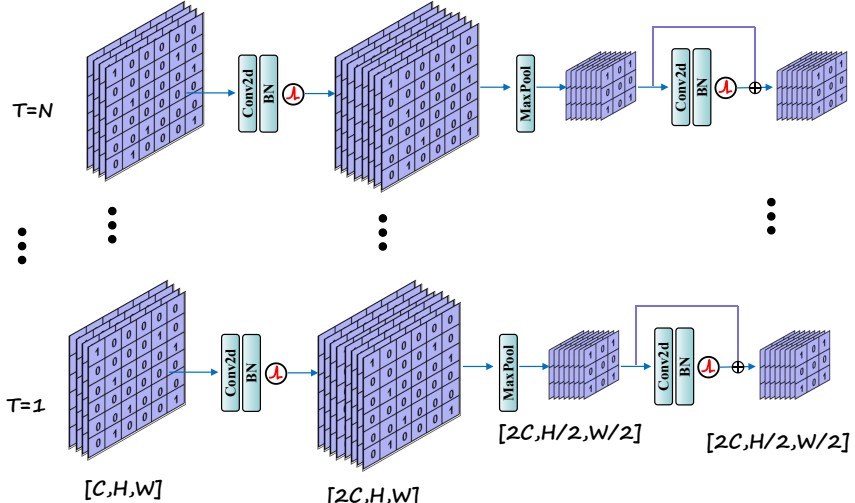

Figure 9: Patch Embedding.

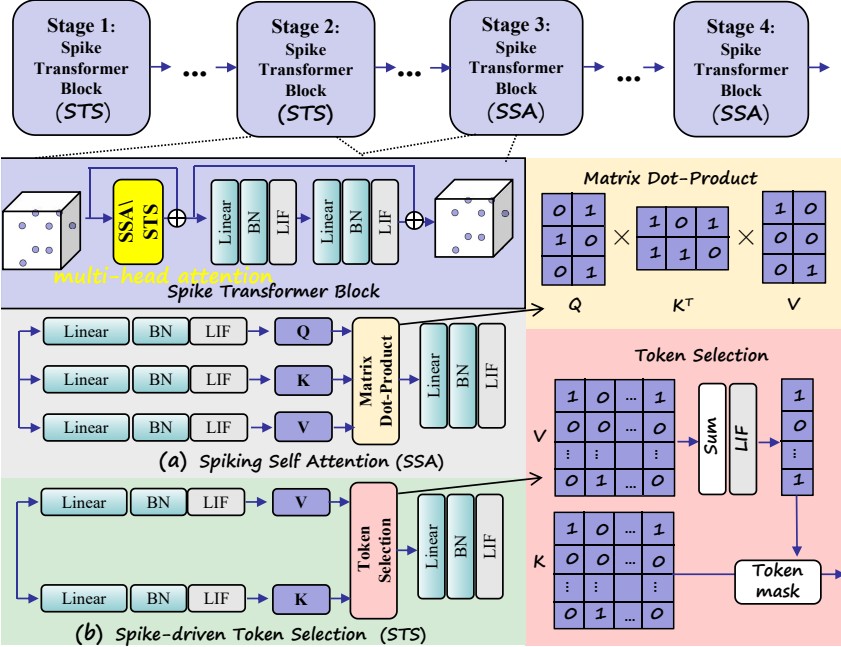

Figure 10: Spike Transformer Block. Spike-driven Token Selecion (STS) is used for token selection in the first two stages, and Spiking Self Attention (SSA) is employed for global feature extraction in the latter two stages. (a) and (b) present the specific details of SSA and QKA. The bottom right corner illustrates the operational processes of 'Matrix Dot-Product' and 'Token Selection' in SSA and QKA, respectively.

$$O^l[t] = A^l[t] \oplus S^l[t], \tag{22}$$

the gradient from the output of the (l+k-1)-th SEW block to the input of the l-th SEW block can be computed in a layer-by-layer manner:

$$\frac{\partial O_j^{l+k-1}[t]}{\partial S_j^l[t]} = \prod_{i=0}^{k-1} \frac{\partial \left( A_j^{l+i}[t] \oplus S_j^{l+i}[t] \right)}{\partial S_j^{l+i}} = \prod_{i=0}^{k-1} \frac{\partial \left( 0 + S_j^{l+i}[t] \right)}{\partial S_j^{l+i}}. \tag{23}$$

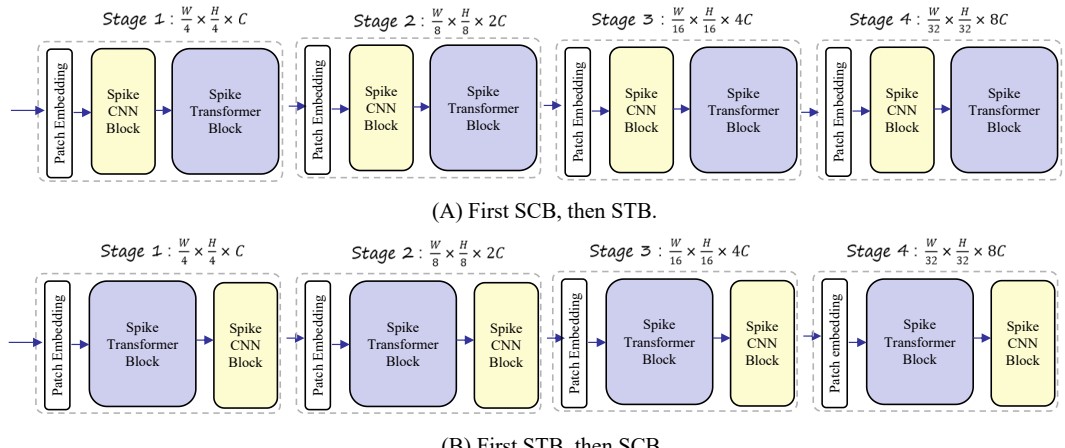

(A) First SCB, then STB.

(B) First STB, then SCB.

Figure 11: The order of Spiking Transformer Block (STB) and Spiking CNN Block (SCB) in the backbone.

Table 7: Performance comparison based on the order of Spiking Transformer Block (STB) and Spiking CNN Block (SCB) in the backbone.

| Backbone | $mAP_L$ | $mAP_M$ | $mAP_S$ | mAP | $mAP_{50}$ |
|---|---|---|---|---|---|
| A | 0.329 | 0.420 | 0.177 | 0.328 | 0.587 |
| B | 0.336 | 0.426 | 0.267 | 0.336 | 0.604 |

The equality holds as identity mapping is achieved by setting $A^{l+i}[t] \equiv 0$. Since the gradient is a constant ( = 1), the SEWA ($\oplus$) can overcome gradient vanishing or gradient explosion.

## A.8 TRAINING DETAILS

In this research, we present an in-depth exploration of the specific training process values for the Gen1 and 1Mpx datasets depicted in Fig. 12 and Fig. 13. These figures not only offer a comprehensive overview of the learning rates and loss values at each iteration but also provide valuable insights into the stability and evolution of the training procedure. Furthermore, the visual representations encapsulate the $mAP_{50}$, $mAP$, $mAP_{large}$, $mAP_{middle}$, and $mAP_{small}$ metrics, affording a holistic comprehension of the network's performance across diverse evaluation criteria throughout the duration of the training process. It is noteworthy that the evaluation metrics consistently demonstrate a gradual enhancement in performance on the training set, underscoring the efficacy and robustness of the training approach adopted in this study. This detailed analysis serves to enrich our understanding of the intricate dynamics and progressive refinement observed during the training phase, thus contributing to a more nuanced interpretation of the network's learning process.

## A.9 MORE VISUALIZATION RESULTS

In addition to the visualization results previously presented, we offer further insights into the characteristics of the Gen1 and 1Mpx datasets in Fig. 14 and Fig. 15, respectively. These visualizations provide a deeper understanding of the dataset attributes and distribution, shedding light on the diverse features and patterns captured within the data.

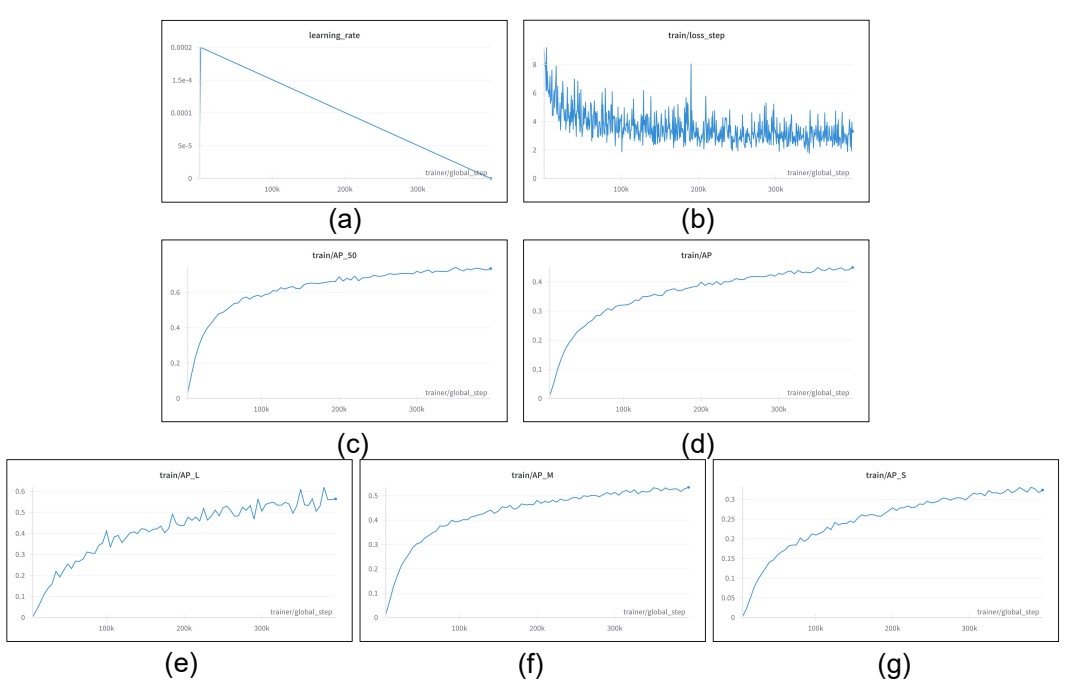

Figure 12: Training Details on the Gen1. (a) Learning rate, (b) loss reduction, (c) $mAP_{50}$, (d)$mAP$, (e) $mAP_{large}$, (f) $mAP_{middle}$, (g) $mAP_{small}$.

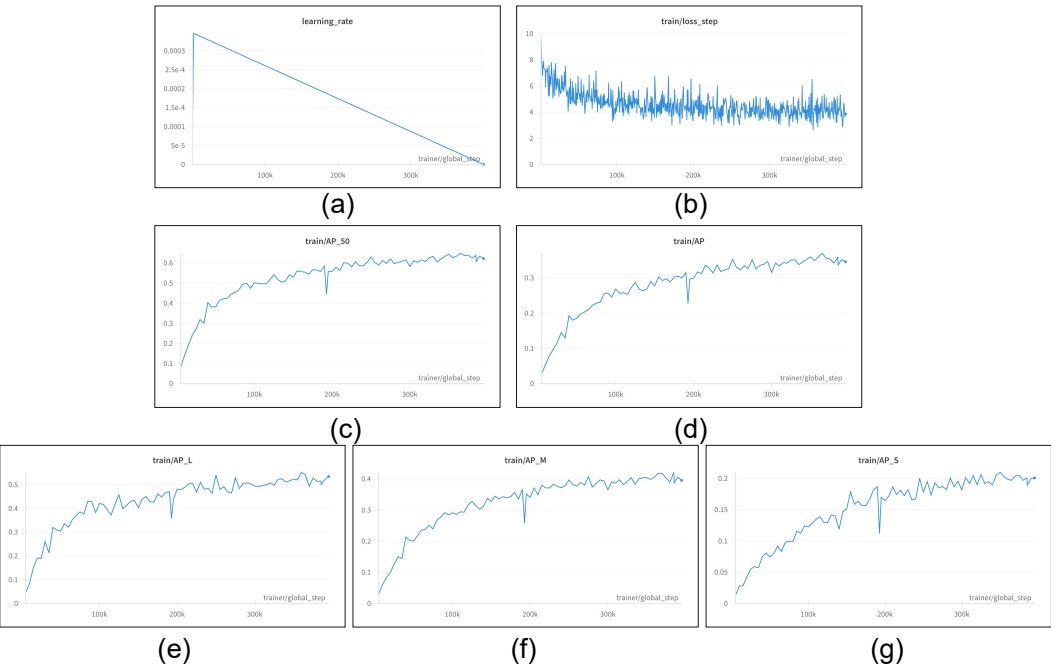

Figure 13: Training Details on the 1Mpx. (a) Learning rate, (b) loss reduction, (c) $mAP_{50}$, (d)$mAP$, (e) $mAP_{large}$, (f) $mAP_{middle}$, (g) $mAP_{small}$.

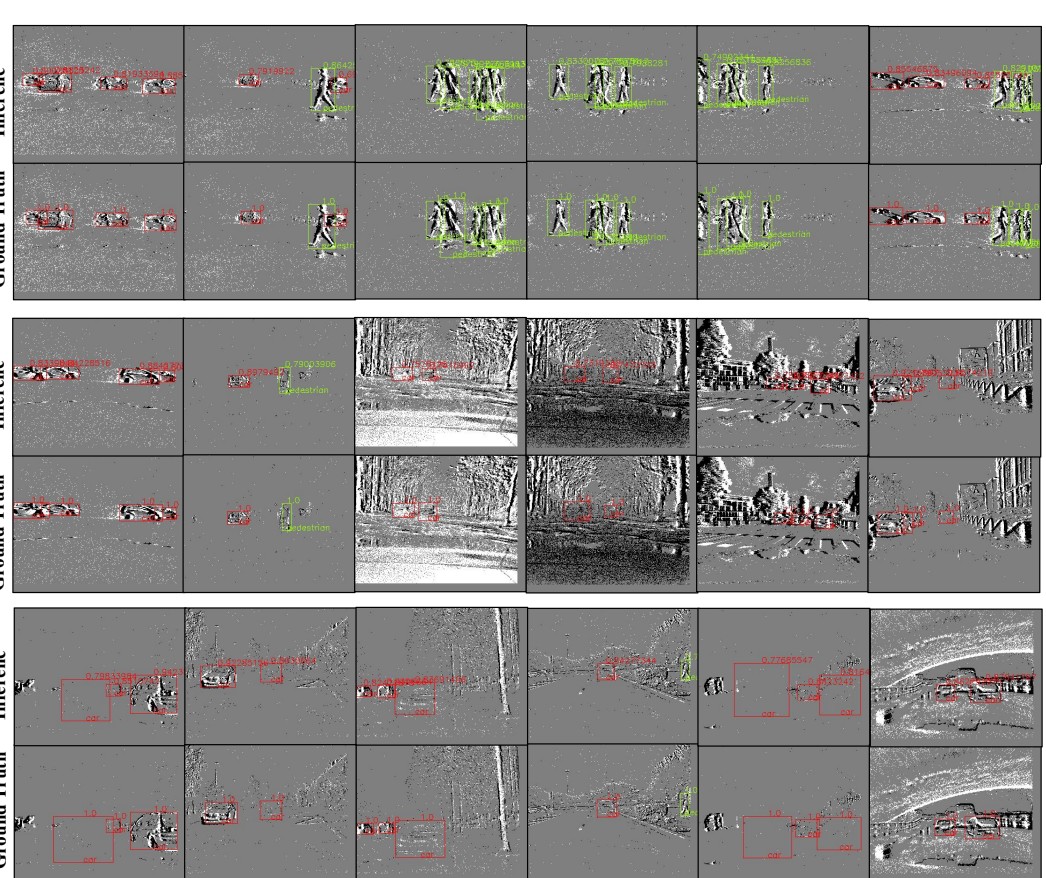

Figure 14: More visualization results on the Gen1 dataset.

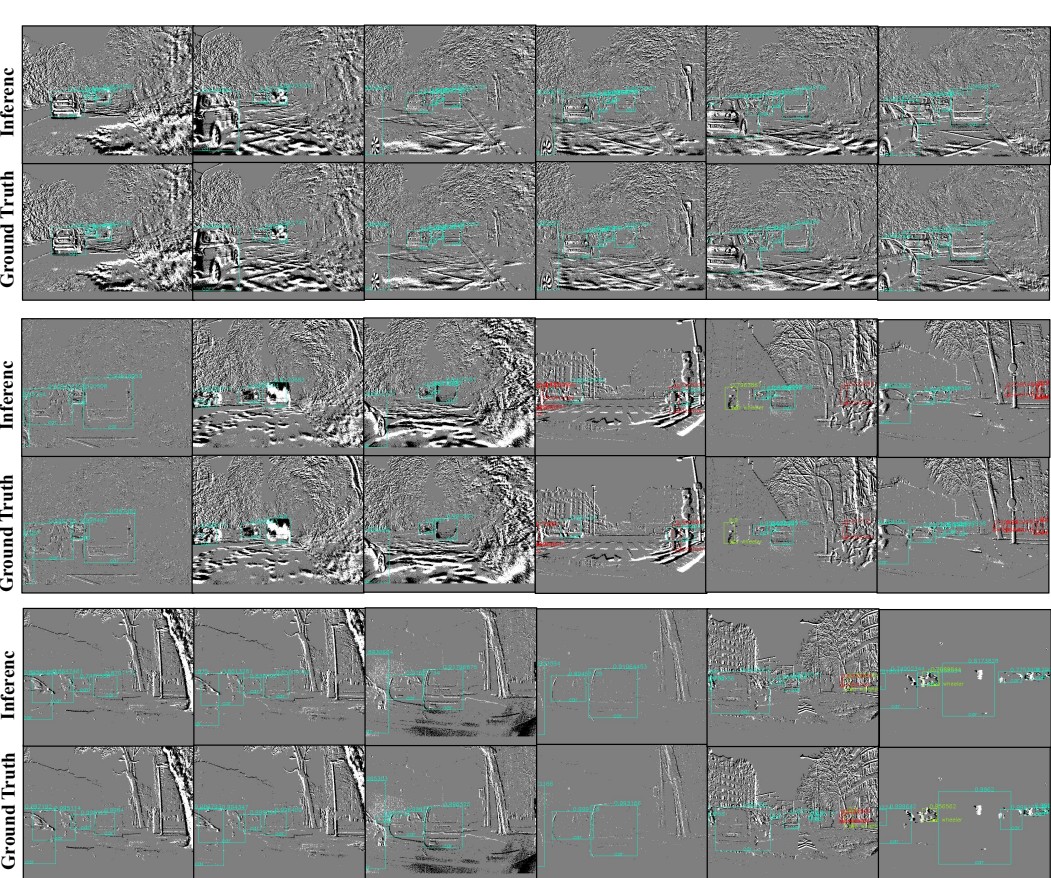

Figure 15: More visualization results on the 1Mpx dataset.

