# OpenReview forum: "Spiking Transformer-CNN for Event-based Object Detection"
_ICLR.cc/2025/Conference — Submitted to ICLR 2025_

### Official Review · Reviewer_RXTy · 2024-10-27

**Soundness:** 2
**Presentation:** 2
**Contribution:** 2
**Rating:** 5
**Confidence:** 4

**Summary:**

The paper proposes a hierarchical Spiking Transformer-CNN that effectively combines global and local information, successfully improving the performance of SNN-based object detectors while reducing power consumption.

**Strengths:**

1. The paper proposes the Spike-TransCNN model, successfully integrating global information from Spiking Transformer with local information from Spiking CNNs, which is beneficial for future development in this field.
2. The paper introduces several interesting blocks, such as STS and SCB, which effectively improve the model's performance.

**Weaknesses:**

1. The integration of global information from Spiking Transformer with local information from Spiking CNNs was first proposed in [1], rather than in this work.
2. The used or proposed modules in this paper, including SSA, SCB, and both intra-stage and inter-stage spike feature fusion modules, fail to preserve the spiking characteristics of SNNs due to their incorporation of non-spiking computations. Consequently, Spike-TransCNN would be more accurately categorized as a hybrid ANN-SNN model rather than a pure SNN.
3. The paper omits comparisons with other pure SNN models (such as SpikeYOLO [2]) and hybrid models (like EAS-SNN [3] and SpikingViT [4]). Furthermore, the model's mAP performance on GEN1 and 1Mpx datasets is substantially inferior to these state-of-the-art approaches.
4. The motivation for proposing STS is unclear - why is it used in shallow layers instead of SSA?
5. The motivation for proposing SCB is also inadequately explained - on lines 260 and 263, it merely states that it "leverages the local feature extraction capabilities" and "captures local multiscale information". However, this raises questions: Couldn't standard 3x3 and 5x5 convolutions achieve the same objective?
6. The reported energy consumption raises significant concerns. Given that Spike-TransCNN has double the parameters of SFOD with similar firing rates, and extensively employs SPIKE-ELEMENT-WISE ADDITION operations (introducing non-spiking computations) across its modules, the claimed lower energy consumption compared to SFOD requires further justification.
7. The paper contains formatting issues, specifically on line 382 where the Firing Rate and Energy (mJ) are overlapping.

[1] Yao M, Hu J K, Hu T, et al. Spike-driven Transformer V2: Meta Spiking Neural Network Architecture Inspiring the Design of Next-generation Neuromorphic Chips[C]//The Twelfth International Conference on Learning Representations.
[2] Luo X, Yao M, Chou Y, et al. Integer-Valued Training and Spike-Driven Inference Spiking Neural Network for High-performance and Energy-efficient Object Detection[J]. arXiv preprint arXiv:2407.20708, 2024.
[3] Wang Z, Wang Z, Li H, et al. EAS-SNN: End-to-End Adaptive Sampling and Representation for Event-based Detection with Recurrent Spiking Neural Networks[J]. arXiv preprint arXiv:2403.12574, 2024.
[4] Yu L, Chen H, Wang Z, et al. Spikingvit: a multi-scale spiking vision transformer model for event-based object detection[J]. IEEE Transactions on Cognitive and Developmental Systems, 2024.

**Questions:**

1. While the paper's primary contribution lies in integrating global information from Spiking Transformer with local information from Spiking CNNs, the motivations for its various modules lack coherent alignment with this central objective.
2. The paper would benefit from additional ablation studies and in-depth analysis of the STS module.
3. The authors are recommended to carefully review the paper for grammatical accuracy and logical coherence.

---

### Official Review · Reviewer_8Cbi · 2024-10-28

**Soundness:** 2
**Presentation:** 2
**Contribution:** 2
**Rating:** 3
**Confidence:** 5

**Summary:**

The paper proposes a novel Spiking Transformer-CNN (Spike-TransCNN) architecture aimed at enhancing event-based object detection by combining the global information extraction capabilities of Spiking Transformers with the local feature extraction strengths of Spiking CNNs. This hybrid approach addresses current limitations in spiking neural networks (SNNs) for object detection, particularly in incorporating both global and multi-scale local features, and demonstrates promising results on the Gen1 dataset.

**Strengths:**

1. The hierarchical integration of Spiking Transformer and CNN blocks is well-motivated and effectively leverages the strengths of each architecture, leading to improvements in both accuracy and energy efficiency.
2. The paper provides robust experimental validation, including comparisons with state-of-the-art methods, and energy efficiency analyses, showcasing the advantages of Spike-TransCNN over conventional ANN-based methods.
3. The focus on energy-efficient event-based object detection aligns with the needs of edge-computing applications, and the results demonstrate significant energy savings, which is a major contribution in neuromorphic computing.

**Weaknesses:**

1. Although the hybrid architecture is described in detail, the explanations of specific processes, such as spike-driven token selection and intra- and inter-stage feature fusion, could be clearer. Including pseudo-code or flow diagrams might enhance the reader’s understanding of the model’s operation.
2. While the results on the Gen1 dataset are compelling, it would strengthen the paper to evaluate the model on more diverse datasets, particularly larger or more complex event-based datasets, to demonstrate generalizability.
3. The paper could benefit from a discussion on how Spike-TransCNN compares with hybrid SNN-ANN models, given their potential to balance energy efficiency and performance. This would contextualize the performance and energy efficiency gains of Spike-TransCNN more effectively.
4. While there is an ablation study on some components, further exploration on the impact of various hyperparameters (e.g., number of time steps, membrane potential thresholds) could provide insights into optimizing the architecture for different applications.
5. Consistent terminology, particularly around spiking mechanisms and attention mechanisms, would improve readability. Some abbreviations and terms could be clarified for non-specialist readers.
6. The paper includes visualization results, but providing side-by-side comparisons with other models on challenging scenarios could offer a clearer view of the model’s strengths in handling occlusions and motion.

**Questions:**

1. Could you provide a more detailed explanation or pseudo-code for the spike-driven token selection and feature fusion processes? It would be helpful to understand the precise mechanics of these operations within the architecture.
2. Have you considered evaluating Spike-TransCNN on additional event-based datasets, possibly with larger or more complex scenes? If so, were there any particular challenges, and if not, could you discuss the potential limitations of the model’s generalizability?
3. How does Spike-TransCNN compare with hybrid SNN-ANN models in terms of both accuracy and energy efficiency? Including such comparisons could help contextualize the advantages of your proposed model.
4. Can you provide more insight into the sensitivity of the model to various hyperparameters, such as the number of time steps or membrane potential thresholds? An ablation study on these parameters might help to optimize the model further.
5. Could you include side-by-side visual comparisons with other models to highlight Spike-TransCNN’s strengths, particularly in scenarios with occlusions or rapid movement? This might better illustrate the advantages of your architecture in challenging conditions.
6. Some of the terms related to spiking mechanisms and attention mechanisms could be more consistently used. Would you consider revising these terms for clarity, especially to make the paper more accessible to readers from a broader audience?

---

### Official Review · Reviewer_bHdm · 2024-10-30

**Soundness:** 2
**Presentation:** 2
**Contribution:** 2
**Rating:** 3
**Confidence:** 4

**Summary:**

This paper proposes Spike-TransCNN, a novel hierarchical architecture combining Spiking Transformers and Spiking Convolutional Neural Networks for event-based object detection. The work addresses the challenge of balancing detection accuracy and energy efficiency in event-based object detection.

**Strengths:**

1. This work is the first attempt to combine Spiking-Transformer and Spiking-CNN architectures for event-based object detection.
2. Spike-TransCNN achieves competitive performance on the Gen1 dataset, whose mAP is 0.336 and energy consumption is 5.49 mJ.
3. The visualization and graphical presentation are of high quality and clarity.

**Weaknesses:**

1. The performance on the 1Mpx dataset (0.250) is significantly inferior to existing methods (0.483), without adequate explanation or analysis.
2. Lack of comprehensive comparisons with recent state-of-the-art SNN detection methods, such as "Integer-Valued Training and Spike-Driven Inference Spiking Neural Network for High-performance and Energy-efficient Object Detection."
3. The whole architecture primarily transplants the established Transformer-CNN paradigm into the SNN domain, with limited Innovation.
4. The paper exhibits an overreliance on descriptive language while lacking theoretical analysis for the performance improvements of the proposed architecture.

**Questions:**

1. In your discussion section, you referenced the "Integer-Valued Training and Spike-Driven Inference Spiking Neural Network for High-performance and Energy-efficient Object Detection." However, no comparative analysis was provided. Could you elaborate on the rationale behind this omission in your experimental evaluation?
2. Regarding the notable performance decline on the 1Mpx dataset, please give an explanation.

---

### Official Review · Reviewer_r8pH · 2024-11-02

**Soundness:** 2
**Presentation:** 2
**Contribution:** 2
**Rating:** 3
**Confidence:** 4

**Summary:**

This article proposes a novel SNN architecture combining Transformer and CNN for object detection. In the analysis process, the author noticed the different roles of Transformer and CNN and effectively combined them. However, in the introduction section, the author's logic seems to focus on describing the development process of SNN, rather than focusing on the problem at hand. Meanwhile, in the narrative of the contribution section, the author's innovative points are not highlighted and appear scattered. In the theoretical/methodological section, the author seems to have referenced previous work and made some iterations. However, the theoretical part did not reflect the author's contribution. This article is not mature and needs to be polished and edited to highlight its own contributions. It must be acknowledged that the author has achieved excellent results on the object detection dataset, however, this seems to depend on the number of module stacks. I suggest rejecting and polishing.

**Strengths:**

This paper proposes a novel hybrid architecture of Transformer and CNN based on pulse neurons, which has achieved good results in event object detection tasks. Among them, the author has made a reasonable design for the Transformer branch and CNN branch under the spikng neuron, and designed a feature fusion method that conforms to the hybrid architecture. The experimental part is relatively complete, with reasonable and clear composition.

**Weaknesses:**

The introduction of this article does not highlight the innovative points. The combination of Transformer and CNN has been proven effective and maturely applied in ANN. The use of LIF neurons for Spiking is not an important innovation point. Furthermore, in the theoretical section, it is evident that both the Transformer architecture and CNN module were designed with reference to previous tasks, which does not constitute an important innovation point to support the paper. The subsequent feature fusion module described two types separately, but the differences between types and the issues addressed were not elaborated in detail. Reading through the methods section, the innovative points and logical description of the methods are similar to the module stacking without highlighting the author's own analysis.

**Questions:**

(1) The Spiking Transformer seems to reference previous work, but its innovation points are not prominent. Has the author made any corresponding contribution?
(2) Can the author explain the issues addressed in the design for the SCB section, which is the CNN module section? And what effect does it have?
(3) Please provide a detailed analysis of the problem solved in the feature fusion section?
(4) Please analyze the overall network design logic. The current analysis tends towards module stacking.

---

### Meta-Review · Area_Chair_KEXT · 2024-12-15

**Metareview:**

The paper presents Spike-TransCNN, a hierarchical architecture combining Spiking Transformers and Spiking CNNs to improve event-based object detection. The proposed method aims to address the limitations of spiking neural networks by integrating both global and multi-scale local features, achieving improved detection accuracy and energy efficiency.

The strengths of this paper include proposing a hybrid Spiking Transformer-CNN architecture for event-based object detection, demonstrating competitive accuracy, energy efficiency, and clear visualizations and experimental results. The weaknesses of this paper include limited novelty, as it primarily adapts existing ANN concepts to SNNs, insufficient comparisons with recent SNN methods, poor performance on larger datasets, and unclear explanations of technical details and module motivations.

This paper received four negative reviews, and the authors did not provide a response. Given this feedback, it is a clear reject.

**Additional Comments On Reviewer Discussion:**

There is no rebuttal or post-rebuttal discussion.

---

### Decision · Program_Chairs · 2025-01-22

Reject